# Rac1/Wave2/Arp3 Pathway Mediates Rat Blood-Brain Barrier Dysfunction under Simulated Microgravity Based on Proteomics Strategy

**DOI:** 10.3390/ijms22105165

**Published:** 2021-05-13

**Authors:** Ranran Yan, Huayan Liu, Fang Lv, Yulin Deng, Yujuan Li

**Affiliations:** School of Life Science, Beijing Institute of Technology, No. 5 Zhongguancun South Street, Haidian District, Beijing 100081, China; yanran12216@163.com (R.Y.); lhy7881230@163.com (H.L.); lvfangbeijing@bit.edu.cn (F.L.); deng@bit.edu.cn (Y.D.)

**Keywords:** simulated microgravity, blood-brain barrier, proteomics, HBMECs, Rac1/Wave2/Arp3

## Abstract

The blood-brain barrier (BBB) is critical to maintaining central nervous system (CNS) homeostasis. However, the effects of microgravity (MG) on the BBB remain unclear. This study aimed to investigate the influence of simulated MG (SMG) on the BBB and explore its potential mechanism using a proteomic approach. Rats were tail-suspended to simulate MG for 21 days. SMG could disrupt the BBB, including increased oxidative stress levels, proinflammatory cytokine levels, and permeability, damaged BBB ultrastructure, and downregulated tight junctions (TJs) and adherens junctions (AJs) protein expression in the rat brain. A total of 554 differentially expressed proteins (DEPs) induced by SMG were determined based on the label-free quantitative proteomic strategy. The bioinformatics analysis suggested that DEPs were mainly enriched in regulating the cell–cell junction and cell–extracellular matrix biological pathways. The inhibited Ras-related C3 botulinum toxin substrate 1 (Rac1)/Wiskott–Aldrich syndrome protein family verprolin-homologous protein 2 (Wave2)/actin-related protein 3 (Arp3) pathway and the decreased ratio of filamentous actin (F-actin) to globular actin contributed to BBB dysfunction induced by SMG. In the human brain microvascular endothelial cell (HBMECs), SMG increased the oxidative stress levels and proinflammatory cytokine levels, promoted apoptosis, and arrested the cell cycle phase. Expression of TJs and AJs proteins were downregulated and the distribution of F-actin was altered in SMG-treated HBMECs. The key role of the Rac1/Wave2/Arp3 pathway in BBB dysfunction was confirmed in HBMECs with a specific Rac1 agonist. This study demonstrated that SMG induced BBB dysfunction and revealed that Rac1/Wave2/Arp3 could be a potential signaling pathway responsible for BBB disruption under SMG. These results might shed a novel light on maintaining astronaut CNS homeostasis during space travel.

## 1. Introduction

During spaceflight, astronauts are exposed to environmental stress, such as microgravity (MG), radiation, and noise. Among these, MG is one of the main hazards threatening human health [1]. MG or simulated MG (SMG) might affect many aspects of human physiology, including cephalad shift of body fluids, loss of fluids and electrolytes [2], bone loss and muscle atrophy [3], and immunological deficiency [4]. In addition, studies have shown that MG could induce a number of neurological disorders, such as learning and memory ability decline [5], cognitive deficits, visual disturbances, movement/orientation control alteration, sleep disorders [6], nausea, and headaches [7]. It is worth noting that cognitive deficits and sleep disorders are associated with blood-brain barrier (BBB) dysfunction, including increased permeability, and downregulated tight junctions (TJs) and adherens junctions (AJs) protein expression [8,9,10]. The BBB consists of the capillary basement membranes, brain microvascular endothelial cell (BMEC), astrocytic end-feet, and pericytes. The major component of the BBB is the BMEC [11]. BBB permeability is mainly determined by two structures: the junction between adjacent BMEC (cell–cell junction) and adhesion of BMEC to extracellular matrix (cell–ECM adhesion) [12]. The cell–cell junction is maintained by TJs and AJs. Furthermore, focal adhesion (FA) composed of adhesion molecules provides an important structural basis for anchoring the endothelial cell–ECM adhesion. TJs are primarily composed of claudins, occludin, and zonula occludens (ZO) proteins. Vascular endothelial cadherin (VE-cadherin) and cytosolic catenins are the central cell–cell junction proteins in AJs [13]. AJs and TJs proteins interact with the actin cytoskeleton [14]. The regulation of the cytoskeleton is involved in the change of BBB permeability [15].

BBB is critical to maintaining CNS homeostasis. The functional status of the CNS is highly related to the astronaut’s physical and psychological health, working efficiency, and safety. Studies on BBB integrity under MG are important to maintain astronaut CNS homeostasis during space travel. Thus far, reports on the integrity of the BBB induced by MG or SMG are hardly available. One study used a mathematical model to predict that exposure to the space environment has an adverse effect on BBB integrity [16]. A recent study demonstrated that spaceflight disturbed mouse BBB integrity through decreased expression of intercellular junction proteins ZO-1 in the mouse brain [17]. It is implied that BBB could be dysfunctional under MG conditions. The existing two reports are far from enough to clarify its inherent mechanisms. Therefore, it is necessary to further study the effects of SMG on the integrity of the BBB and explore its potential etiological mechanism.

This study aimed to investigate the effects of SMG on the integrity of the BBB and explore its potential etiological mechanism based on a proteomic strategy. Tail-suspended rats (also known as the Morey–Holton model, approved by the National Aeronautics and Space Administration) were used to simulate MG for 21 days [18]. The effects of SMG on the BBB were investigated via histomorphology and ultrastructure examination. BBB permeability was examined by the extravasation of classical tracer Evans blue (EB) and Texas red-dextran (Dex). BBB damage induced by SMG was evaluated by assessing the oxidative stress levels and proinflammatory cytokine levels and TJs and AJs protein expression in the rat brain. Label-free proteomics was employed to profile the overall changes of SMG rat brain proteins and further explore the potential pathway related to BBB dysfunction induced by SMG. The potential signaling pathway of BBB dysfunction induced by SMG was validated in rats. The key role of the potential signaling pathway in BBB damage was further confirmed in human brain microvascular endothelial cells (HBMECs) with a specific agonist. The potential mechanism of BBB dysfunction by SMG was first elucidated. The expected findings might help to understand the response of BBB to SMG and provide a pathological and mechanical basis to manage the BBB integrity of astronauts during space missions.

## 2. Results

### 2.1. 21-Day SMG (21d-SMG) Damaged Brain Histomorphology and BBB Ultrastructure

The hematoxylin and eosin (H&E) staining results are shown in Figure 1A. Compared to the histomorphological image of CON, the SMG group showed signs of inflammatory cellular infiltration and nuclear pyknosis in the cerebral cortex (green arrows). The BBB ultrastructure observed with transmission electron microscopy (TEM) is shown in Figure 1B. In the cerebral cortex, the SMG group showed a widened intercellular space (red arrows). Meanwhile, there were observed swollen pericytes (green arrows) and unclear mitochondria cristae (blue arrows). Collectively, these results indicated that 21d-SMG damages brain histomorphology and BBB ultrastructure.

### 2.2. 21d-SMG Induced BBB Hyperpermeability in Rats

The spectrophotometric determination of EB and Dex stain accumulation in the brain is easy, reliable, and the most widely used technique to estimate BBB permeability [19,20]. The effects of SMG on BBB permeability were assessed by the extravasation of EB and Dex. As shown in Figure 2, compared to the CON group, the EB and Dex content in the rat brains of the SMG group was significantly increased by 31.1% (*p* < 0.05) and 37.2% (*p* < 0.01), respectively. The result suggested that SMG significantly induces increased rat BBB permeability.

### 2.3. 21d-SMG Increased Oxidative Stress Levels and Proinflammatory Cytokine Levels in the Rat Brain

Compared to the CON group, tumor necrosis factor-α (TNF-α), interferon-γ (IFN-γ), and interleukin-6 (IL-6) in the SMG group were significantly increased by 29.7%, 38.9%, and 21.6%, respectively (*p* < 0.05; Figure 3A). The increase in pro-inflammatory cytokine levels might indicate that SMG induced inflammatory injury in the rat brain. In addition, the malondialdehyde (MDA) and levels and hydrogen peroxide (H_2_O_2_) content and superoxide dismutase (SOD) activity in the rat brain were assessed. Compared to the CON group, the MDA and H_2_O_2_ content in the rat brain was remarkably increased by 50.2% and 37.8%, respectively (*p* < 0.05). The SOD activity in the rat brain was decreased by 24.5% in the SMG group (*p* < 0.05; Figure 3B). These results suggested that SMG could increase rat brain oxidative stress levels and proinflammatory cytokine levels.

### 2.4. SMG Downregulated TJs and AJs Proteins in the Rat Brain

Western blot analysis (Figure 4) shows that claudin-5, VE-cadherin, and β-catenin expression dramatically decreased by 51.1%, 47.6%, and 63.1%, respectively (*p* < 0.05) under SMG conditions. There was no significant difference in occludin and ZO-1 expression. Immunohistochemistry (IHC) staining was used to investigate the expression and distribution of claudin-5, occludin, and ZO-1. The results revealed that claudin-5, occludin, and ZO-1 expression was consistent with the detection of TJs protein expression by Western blot analysis (Appendix A). 21d-SMG did not alter the distribution of TJs protein in the rat brain. These results suggested that SMG significantly downregulates claudin-5, VE-cadherin, and β-catenin expression by Western blot and IHC analyses.

### 2.5. Proteomics Analysis

A total of 1295 proteins that were identified in the rat brain are shown in the volcano plot in Figure 5A. DEPs were selected using the following conditions: *p* < 0.05 and fold change cutoff was set to >1.5 and <0.667 for upregulation and downregulation, respectively. A total of 554 proteins were identified as DEPs. Among them, 464 DEPs were downregulated and 90 DEPs were upregulated. The detailed information for these 544 proteins is shown in Appendix A. Focal adhesion kinase (Ptk2) downregulation was examined using Western blot analysis to confirm data from the MS assay (Figure 5B). The proteomic data were further analyzed by the bioinformatics tool DAVID version 6.8. The functions of DEPs were annotated according to GO, and the KEGG pathway was analyzed.

#### 2.5.1. GO Analysis

The GO cluster of DEPs provided a preliminary description of the potential functions. The GO terms with *p* < 0.01 were annotated into eight clusters (enrichment scores ≥ 2.61). The top four clusters are listed in Table 1. Most DEPs were enriched with cell adhesion, proteasome complex, translation, and small GTPase-mediated signal transduction. Cell adhesion exhibited the highest enrichment score of 13.61. The 36 DEPs related to cell adhesion are listed in Table 2. Among DEPs, 30 proteins were downregulated, such as vinculin, fascin, and Rab (1a, 10, and 11b). Cell adhesion can be divided into cell–cell and cell–ECM adhesion. The adhesion of BMECs to ECM is also an important determinant of BBB permeability. Cell–ECM adhesion ensures the morphology and function of cerebral endothelial cells and maintains BBB integrity. Cell–ECM adhesion mainly depends on FA complexes that bind endothelial cells to ECM. FA are multiprotein complexes composed of vinculin, Rab1a, the FA component p130Cas, Crk, and so on [21,22]. These results showed that vinculin, Rab1a, and fascin were downregulated 0.06 to 0.43 times the CON group.

GO analysis showed that 82.9% of DEPs involved in cell adhesion were downregulated under SMG. SMG suppressed the expression of adhesion molecules, for example, vinculin, fascin, Rab1a, etc. The downregulation of adhesion molecules suggested that SMG might induce the impairment of cell–ECM and cell–cell adhesion. Cell–cell adhesion is primarily maintained by TJs and AJs. Claudin-5 and β-catenin are considered as the key adhesion molecules forming TJs and AJs structures, respectively. The damaged cell–cell adhesion found in GO analysis was in concordance with the downregulation of claudin-5 and β-catenin expression detected by the Western blot method. Collectively, SMG might impair cell adhesion structures, thereby increasing BBB permeability. The downregulated vinculin, fascin, and Rab1a suggested that SMG might play a key role in damaging cell adhesion structures. The cell adhesion impairment might induce BBB hyperpermeability.

#### 2.5.2. KEGG Pathway Analysis

The KEGG pathway analysis is shown in Figure 5C. In total, 26 of the main pathways were analyzed, and the top one was metabolic pathways, including amino acids, proteins, and carbon metabolism. The following three pathways were biosynthesis of antibiotics, endocytosis, and dopaminergic synapse. In addition, some DEPs were enriched with the disease pathways, including Huntington’s, Parkinson’s, and Alzheimer’s diseases. Aside from the above pathways, the other pathways were associated with cell adhesion, including regulation of actin cytoskeleton, bacterial invasion of epithelial cells, and focal adhesion. Differently expressed proteins involved in the regulation of actin cytoskeleton and bacterial invasion of epithelial cells are listed in Table 3.

The regulation of the actin cytoskeleton is important for cell adhesion maintenance. In endothelial cells, branched actin polymerization consequently induces the formation of lamellipodia and filopodia protrusion. Lamellipodia and filopodia protrusion is important for FA formation [23,24,25]. FA provide an important structural basis for cell–ECM adhesion [25]. Lamellipodia and filopodia protrusion is also implicated in the maintenance of the cadherin/catenin complex-mediated endothelial cell–cell junction and integrity [26,27]. Bacterial invasion of epithelial cells could deteriorate the endothelium barrier integrity by disrupting TJs and AJs [28]. Based on the GO analysis results, the influence of the cell adhesion-related pathway on BBB permeability was further analyzed. Bioinformatics analysis indicated that actin-related proteins like and actin-related protein 1 (Arp1), actin-related protein 2 (Arp2), and actin-related protein 5 (Arp5) in the SMG group were significantly decreased by 90.19% 98.7%, and 59.73%, respectively. Meanwhile, the expression of Ras-related C3 botulinum toxin substrate 1 (Rac1) was dramatically reduced by 37% under 21d SMG. It has been reported that Arp2/3 is a key regulator of lamellipodia and filopodia protrusion formation by initiating branched actin polymerization [29]. The Rac1/Wiskott–Aldrich syndrome protein family verprolin-homologous protein 2 (Wave2)/Arp3 signaling pathway has been identified as involved in branched actin polymerization at the cell leading edge and the consequent formation of lamellipodia and filopodia [30]. Lamellipodia and filopodia protrusion is implicated in the maintenance of the endothelial cell–cell junction and cell–ECM adhesion. Thus, it was proposed that the Rac1/Wave2/Arp3 signaling pathway may regulate the BBB integrity under SMG.

### 2.6. SMG Inhibited the Rac1/Wave2/Arp3 Pathway

Rac1, Wave2, and Arp3 expression in the rat brain were detected by Western blot analysis (Figure 6). Compared to the CON group, Rac1 and Arp3 in the SMG group were significantly decreased by 36.4% and 56.1%, respectively (*p* < 0.05). Wave2 in the SMG group was downregulated. Consistent with the downregulation of Rac1, Wave2, and Arp3 expression, the ratio of filamentous actin (F-actin) to globular actin (G-actin) was also markedly reduced (*p* < 0.05). Previous studies demonstrated that inhibiting the Rac1/Wave2/Arp2/3 signaling pathway in breast cancer cells and glioma cells resulted in the damage of cell adhesion [31]. In summary, SMG might reduce branched actin polymerization through the Rac1/Wave2/Arp3 pathway, damage cell adhesion, and eventually cause BBB hyperpermeability.

### 2.7. SMG Induced Apoptosis and Oxidative Stress Injury and F-Actin Distribution in HBMECs

Annexin V conjugated to FITC and flow cytometry was used to detect the effects of SMG on cell apoptosis. After SMG for 24 h, the percentage of early and late apoptotic cells was higher among HBMECs in the SMG group than in the CON group (*p* < 0.05). Figure 7A shows the representative results of apoptosis analyzed by flow cytometry and the quantitative comparison results. Flow cytometry analysis showed that the percentages of HBMECs in the G2/M phase were higher in the SMG group (23.03%) than in the CON group (13.96%; *p* < 0.05; Figure 7B). These results suggested that SMG inhibits HBMECs cycle progression and arrests HBMECs at the G2/M phase of the cell cycle. SMG arrested mouse skeletal myoblasts at the G2/M phase of the cell cycle [32]. To investigate the effects of SMG on the stress level in HBMECs, the effects of SMG on the MDA and H_2_O_2_ content and the SOD activity in HBMECs were investigated. Compared to the CON group, the MDA content and H_2_O_2_ levels were increased remarkably, and the SOD activity decreased in the SMG group. Consequently, SMG could remarkably increase the oxidative stress in HBMECs (Figure 7C). The effects of SMG on proinflammatory cytokine levels were investigated by assessing IL-6, IFN-γ, and TNF-α levels. The proinflammatory cytokine (IL-6, IFN-γ, and TNF-α) levels in HBMECs were upregulated (Figure 7D). Oxidative stress and inflammatory injury are two important contributions to BBB dysfunction. The BMEC barrier integrity is differentially influenced by the localization change in F-actin [33]. The CON group showed that F-actin was spindle distributed in the periphery of the cell nucleus, and the fluorescent staining was bright. Although the fluorescence of F-actin fibers in the SMG group was disordered, the morphology of the fibers was obviously changed, and the fluorescence staining decreased obviously (Figure 7E). SMG altered the distribution of F-actin myofilament in HBMECs, which might damage the BMEC barrier integrity.

### 2.8. SMG Downregulated AJs and TJs Protein Expression and the Rac1/Wave2/Arp3 Pathway in HBMECs

Figure 8A and Appendix A show that SMG dramatically downregulated TJs and AJs protein expression in HBMECs; for example, occludin was the most significantly decreased by 78.7%, and ZO-1, claudin-5, VE-cadherin, and β-catenin were decreased by 24.5% to 64.8%. The downregulation of claudin-5, VE-cadherin, and β-catenin in HBMECs was consistent with the above findings in the rat brain. Western blot assay showed that Rac1, Wave2, and Arp3 expression was dramatically decreased by 26.0%, 26.9%, and 33.3% in the SMG group compared to the CON group (*p* < 0.05). The downregulation of Rac1, Wave2, and Arp3 expression indicated that the Rac1/Wave2/Arp3 signaling pathway in HBMECs was inhibited. Moreover, the ratio of F-actin to G-actin was also markedly reduced (*p* < 0.05), suggesting that SMG inhibits HBMECs actin polymerization (Figure 8B). To further validate the effects of SMG on the Rac1/Wave2/Arp3 pathway, HBMECs were pretreated with a specific Rac1 agonist (O-Me-cAMP). The results suggested that downregulated Rac1, Wave2, and Arp3 expression and decreased ratio of F-actin to G-actin in SMG-treated HBMECs were largely diminished by the agonist (Figure 8C; Appendix A). Therefore, the Rac1/Wave2/Arp3 pathway played a crucial role in BBB dysfunction under SMG.

## 3. Discussion

The H&E staining results suggested that SMG induced inflammatory injury in the rat cerebral cortex. An inflammatory injury would increase BBB permeability, by modifying the localization of TJs proteins and downregulating the expression of TJs proteins [34]. Also, the intercellular space was widened under SMG. The widened intercellular space would induce BBB hyperpermeability. Pericyte is a crucial cellular element of the BBB, interacting with the other components of the BBB for maintaining proper BBB function [35]. An available study found that the ischemic stroke model caused pericyte ultrastructure degeneration and rat BBB hyperpermeability [36]. In addition, the destruction of mitochondria cristae revealed that SMG might have a change of oxidative stress levels and spontaneously induce BBB dysfunction [37]. 21d-SMG induced BBB hyperpermeability in rats. The increase of BBB permeability would allow inflammatory factors to enter the brain parenchyma, causing neuroinflammation [38]. In the present study, H&E staining result indeed suggested that SMG induced inflammatory injury in rat cerebral cortex. In addition, the destruction of intercellular adhesion is related to BBB hyperpermeability. The presence of widened intercellular space observed by TEM was consistent with the increase of BBB permeability under SMG conditions. 21d-SMG increased oxidative stress levels and proinflammatory cytokine levels in the rat brain. An inflammatory injury would increase BBB permeability [34]. Also, 21d-SMG induced inflammatory infiltration as observed by H&E staining. The increased proinflammatory cytokine levels and H&E staining observations further illustrated that 21d-SMG might lead to BBB hyperpermeability. Spaceflight induces oxidative damage in numerous tissues, for example, in the brain, liver, ocular tissue, bone, etc. [39,40]. Ground-based simulations of an MG study also showed that SMG could increase oxidative stress in the rat brain [41,42]. Oxidative stress is considered an important contributor to BBB dysfunction. Oxidative stress activates MMP-9, which contributes to the breakdown of ECM and directly degrades TJs proteins, leading to BBB damage [43]. One study suggested that the increased MDA levels induced a dramatic reduction of TJs-associated proteins, thereby disrupting rat BBB [44]. Another study discovered that increased MDA content and decreased SOD activity in rat brain tissue significantly induced BBB hyperpermeability in the rat ischemia-reperfusion injury model [45]. The increase in the H_2_O_2_ level induced BBB hyperpermeability in the rat model of high-altitude cerebral edema [46]. Collectively, the increased oxidative stress levels and proinflammatory cytokine levels might degrade ECM and TJs proteins and then induce BBB damage.

It is known that TJs and AJs proteins are important proteins that compose BBB. In the claudin family proteins, claudin-5 mainly participates in the formation of TJs [47]. Claudin-5 downregulation would increase BBB permeability in neurodegenerative disorders, such as Alzheimer’s disease and depression [48,49]. VE-cadherin and β-catenin are involved in forming AJs protein complexes. One study reported that inflammatory factors affect the stability of BBB by decreasing VE-cadherin in rats [50]. Anesthesia surgery downregulated β-catenin expression and increased BBB permeability [51]. Our results showed that SMG downregulated claudin-5, VE-cadherin, and β-catenin expression which might impair TJs and AJs structures. TJs and AJs structure malfunction might have an influence on the junction between adjacent BMEC and then, might induce the dysfunction of BBB. Downregulation of claudin-5, VE-cadherin, and β-catenin expression was consistent with widened endothelial intercellular space, observed in cerebral cortex′ ultrastructure. In addition, oxidative stress levels and inflammatory factors contribute to degrade TJ and AJ proteins and then damage BBB. The present study found that oxidative stress levels and several inflammatory factors were increased, which may have an adverse effect on BBB integrity.

Proteomics results showed that as adhesion molecules, vinculin, Rab1a, and fascin were downregulated to 0.06 to 0.43 times of the CON group. Vinculin is a kind of cytoplasmic linker protein and could stabilize cell–ECM adhesions by orchestrating the recruitment and release of other cell–ECM proteins [52,53,54]. Vinculin also directly binds the Arp2/3 complex to promote lamellipodial protrusion [55]. Lamellipodia and filopodia protrusions are important for FA formation [56,57,58]. FA provide an important structural basis for cell–ECM adhesions. The destruction of cell–ECM adhesion would trigger the increase of BBB permeability. Downregulated vinculin affected BBB permeability by damaging the adhesion structure in the mice brain model of experimental autoimmune encephalomyelitis [59]. A recent study showed that real spaceflight induced the downregulation of vinculin expression and attenuated cell adhesion in human breast cancer cells [60]. In this study, vinculin expression was downregulated in the SMG rat brain, suggesting that downregulated vinculin damages adhesion structure integrity and eventually increases BBB permeability under SMG.

Fascin is an actin-binding protein involved in cell adhesion through the formation of filopodia [61]. Fascin could recruit the branched actin filaments produced by the Arp2/3 complex to form bundled filopodia [62]. The knockdown of fascin in human colon carcinoma cells leads to the disassembly of FA [63]. Previous reports also showed that fascin downregulation would disrupt cell–ECM contact in neoplasm cells [64]. The small G-protein Rab1a is an important mediator of integrin inside-out signaling, which plays a pivotal role in cell–ECM adhesion [65]. Upon activation, Rab1a has the ability to increase the affinity of integrins for their ECM ligands and promote cell–ECM adhesion [64]. Recent reports have shown that knockdown of Rab1a in prostate cancer cells suppresses cell adhesion [65]. Fascin and Rab1a expression in the rat brain under MG and SMG was not reported, but it decreased significantly in this study. The downregulated fascin and Rab1a suggested that SMG might play a key role in damaging cell adhesion structures. The cell adhesion impairment might induce BBB hyperpermeability.

Bioinformatics analysis indicated that actin-related proteins like Arp1, Arp2, and Arp5 in the SMG group were significantly decreased. Meanwhile, the expression of Rac1 dramatically reduced under 21d SMG. It has been reported that Arp2/3 is a key regulator of lamellipodia and filopodia protrusion formation by initiating branched actin polymerization [66]. Actin polymerization levels can be expressed by an F-actin-to-G-actin ratio. The inhibition of branched actin polymerization means depolymerizing F-actin into G-actin [67]. Three- and 10-day cultures of EA.hy926 endothelial cells in space induced a significant decrease in the amount of F-actin [68]. The Arp2/3 complex requires activation by nucleation promotion factors (NPFs). The most important NPFs belong to Wave2 [66]. Wave2 is activated at the plasma membrane by Rac1 and then engages the Arp2/3 complex to generate a dendritic array of branched filaments responsible for the protrusion of lamellipodia and filopodia [66]. The Rac1/Wave2/Arp3 signaling pathway has been identified as involved in branched actin polymerization at the cell leading edge and the consequent formation of lamellipodia and filopodia. Lamellipodia and filopodia protrusion is implicated in the maintenance of the endothelial cell–cell junction and cell–ECM adhesion.

Arp3 expression was significantly decreased when Xenopus laevis embryos were grown in a random positioning machine [69]. Research also revealed that clinostat-modeled SMG downregulated Rac1 expression in BL6-10 melanoma cells [70]. In a study related to Drosophila melanogaster BBB, Arp2/3 complex knockdown leads to discontinuity in the appearance of septate junctions and BBB hyperpermeability [71]. In Sertoli cells, the downregulation of Rac1 expression induced actin architecture disruption, which destructed the conspicuous dysfunction of TJs proteins at the cell–cell interface, leading to blood-testis barrier hyperpermeability [72]. The available reports indicate that Rac1/Wave2/Arp3 signaling pathway plays an important role in the junction between adjacent BMECs and the adhesion of BMECs to ECM, causing BBB hyperpermeability. The results of the widened intercellular space observed by TEM and downregulated TJs and AJs proteins might implicate that cell adhesion was damaged and the current study also detected that BBB permeability was increased. In addition, the results of the KEGG pathway analysis and GO analysis were similar. Both results indicated that DEPs were mainly enriched in cell adhesion. Furthermore, bioinformatics analysis showed that the pivotal proteins that were involved in the Rac1/Wave2/Arp3 signaling pathway were downregulated under SMG. Also, the key role of the Rac1/Wave2/Arp3 pathway in BBB dysfunction was confirmed in HBMECs with a specific Rac1 agonist. In summary, this study demonstrated that SMG induced BBB dysfunction and revealed that Rac1/Wave2/Arp3 could be a potential signaling pathway responsible for BBB disruption under SMG. These results might shed a novel light on maintaining astronaut CNS homeostasis during space travel.

## 4. Materials and Methods

### 4.1. Animal Treatment and Sample Collection

All animal experiments were conducted in accordance with the Guide for the Care and Use of Laboratory Animals published by the National Institutes of Health (NIH publication no. 85–23, revised in 1996), and all animal experiments were approved by the Beijing Institute of Technology Animal Care and Use Committee (Beijing, China). The approval number was 2018-0003-M-2020009 (approved in April 2020). Fifty-four male Sprague-Dawley rats (220 ± 20 g, 10 weeks old, SPF degree) were purchased from the Academy of Military Medical Sciences (Beijing, China). Rats were kept in a temperature, humidity, and illumination-controlled room (temperature: 24 °C ± 1 °C, humidity: 55% ± 5%, and illumination: 12 h light/dark cycle) with free access to water and normal standard chow diet. All animals were housed for one week before the study. Rats were then randomly divided into six groups (nine rats per group): three control groups (CON I, II, and III), rats were freely raised in normal cages; three SMG groups (SMG I, II, and III), rats were tail-suspended for 21 days. A tail suspension was performed according to Morey–Holton’s protocol [18]. Briefly, disinfected rat tail was attached with a surgical tape and then connected to a pulley by a metal bar. The hindlimbs of rats were elevated through the suspended tail to produce a −30° tilt angle in relation to the horizontal. The rat could freely move in any direction with its forelimbs and had free access to water and food. Rats from CON I and II and SMG I and II were applied for BBB permeability measurement. These rats were injected intravenously with EB or Dex, and rat brain samples were collected. Rats from CON III and SMG III were euthanized by cervical dislocation. After the brain samples were collected, half of the brain samples were used for further histomorphology and ultrastructure experiments. The other half of the brain samples were kept at −80 °C and subsequently applied to determine oxidative stress levels and proinflammatory cytokine levels, TJs and AJs protein expression, and proteomic analysis.

### 4.2. Measurement of BBB Permeability

EB (69 kDa) and Dex (3 kDa) were used as tracers to evaluate BBB permeability following previously described methods. Briefly, 80 mg/kg EB (Sigma, MO, USA) and 7.5 mg/kg Dex (Thermo Fisher Scientific, MA, USA) were injected intravenously into rats and allowed to circulate for 60 min. Rats were then anesthetized with 30 mg/kg sodium pentobarbital and transcardially perfused with saline. After sacrifice, rat brain samples were collected. For EB extravasation analysis, 1 g rat brain sample was mixed with 4 mL trichloroacetic acid and then homogenized. The homogenate was centrifuged (10,000 rpm, 20 min) at 4 °C, and the supernatant was collected immediately. The absorbance of the supernatant was measured at 632 nm with a multifunctional enzyme marker (Thermo Fisher Scientific). For Dex extravasation analysis, 1 g rat brain sample was mixed with 4 mL Tris-HCl (pH 7.6) and then homogenized. The homogenate was mixed with methanol (1:1, *v*/*v*) and centrifuged (10,000 rpm, 10 min) at 4 °C. The fluorescence intensity of the supernatant was measured at excitation and emission wavelengths of 493 and 520 nm, respectively. The EB or Dex content in each sample was calculated from its standard curve.

### 4.3. Histomorphology and Ultrastructure Observation of Rat Brain

#### 4.3.1. The H&E Staining

The cerebral cortex isolated from the rat brain was fixed in 4% paraformaldehyde (Solarbio, Beijing, China). The fixed cerebral cortex was dehydrated with various concentrations of xylene and ethanol and embedded in paraffin [73]. The 4 μm sections were sliced and stained with H&E. The cerebral cortex sections were scanned using a Nano zoomer S210 microscopic resolution scanner (Hamamatsu Corporation, Hamamatsu, Japan).

#### 4.3.2. Transmission Electron Microscopy (TEM) Observation

The BBB ultrastructure was detected by TEM. Briefly, rat brain tissue (1 mm^3^) was fixed for 2 h at 4 °C with 2.5% cold glutaraldehyde and soaked in 2% osmium tetroxide for 2 h in sequence. Then, the samples were washed and dehydrated. Ultrathin sections were prepared and stained with uranyl acetate and lead citrate [74]. Finally, the ultrastructure changes of the BBB were examined with a TEM (Japan Electron Optics Laboratory, Tokyo, Japan).

### 4.4. Cell Culture and MG Simulation

HBMECs were purchased from the ICellBioscienceInc Corporation (Shanghai, China) and cultured in RPMI-1640 (Gibco, CA, USA) medium supplemented with 10% fetal bovine serum (Biological Industries, CA, USA) at 5% CO_2_ atmosphere at 37 °C.

HBMECs were seeded in a T-25 flask (2 × 105 cells/flask). The T-25 flask was carefully filled with culture medium (ensuring no bubbles to avoid shearing of the fluid). HBMECs were cultured in a 3D clinostat (<10^−3^ G; National Space Science Center, Beijing, China) to simulate MG for 24 h [75]. The control group was treated at 1 G in the same CO_2_ incubator. The cell experiment was applied to find the effects of SMG on HBMECs, including cell apoptosis assays, cycle analysis, oxidative stress levels, and proinflammatory cytokine levels. This study also detected the expression of TJs and AJs proteins and signaling pathway proteins with an agonist.

#### Cell Apoptosis Assays and Cell Cycle Analysis

SMG-treated HBMECs pellets were collected by centrifugation and washed twice with phosphate-buffered saline (PBS). Aliquots of 1 × 10^6^ cells were suspended in 100 µL binding buffer containing fluorescein isothiocyanate (FITC)-conjugated Annexin V and propidium iodide (PI). After incubation in the dark and on ice for 15 min, cells were analyzed using a flow cytometer (Beckman Coulter, CA, USA) [76]. The cell cycle proportion of HBMECs was carried out using flow cytometric analysis as described previously. Briefly, SMG-treated cells were fixed with 70% ethanol overnight and then stained with PI in the dark for 20 min. Eventually, the DNA content was detected by a flow cytometer.

### 4.5. Detection of Oxidative Stress Levels and Proinflammatory Cytokine Levels in HBMECs and Rat Brain

The brain and HBMECs samples were collected and homogenized in a radioimmunoprecipitation assay (RIPA) lysis buffer (Roche, Basel, Switzerland). The brain and HBMECs homogenates were centrifuged at 12,000× *g* for 10 min at 4 °C, and the protein concentration of the supernatant was determined. The level of hydrogen peroxide (H_2_O_2_) and malondialdehyde (MDA) were estimated with a Hydrogen Peroxide Assay Kit (Cat# A003-1-1, Nanjing Jiancheng Bioengineering Institute, Nanjing, China) and the Lipid Peroxidation (MDA) Assay Kit (Cat# A007-1-1, Nanjing Jiancheng Bioengineering Institute, Nanjing, China), all according to the manufacturer’s instructions. Superoxide Dismutase (SOD) activity was determined with the Superoxide Dismutase (SOD) Colorimetric Activity Kit (Cat# A001-3-2, Nanjing Jiancheng Bioengineering Institute, Nanjing, China) according to the manufacturer’s instructions. Briefly, the level of MDA was determined by the thiobarbituric acid reactive substances method (TBARS). The method was used to obtain a spectrophotometric measurement of the color produced during the reaction of TBA with MDA at 535 nm. SOD activity was measured following the reduction of nitrite by a xanthine–xanthine oxidase system which is a superoxide anion generator. The level of hydrogen peroxide (H_2_O_2_) was determined by the complex of Hydrogen peroxide and molybdate acid at 405 nm.

### 4.6. Western-Blot Analysis

Rat brain tissue and HBMECs samples were homogenized in a precooled RIPA lysis buffer containing protease and phosphatase inhibitors. The suspension was centrifuged at 12,000× *g* for 10 min at 4 °C, and the supernatant was collected immediately. The protein concentration of each sample was determined by the Bradford method (Bio-Rad, CA, USA). Equal amounts (30 μg) of protein for each sample were separated by sodium dodecyl sulfate-polyacrylamide gel electrophoresis (SDS-PAGE; 5% stacking gel, 80 V for 20 min, and 12% separating gel, 110 V for 80 min) and then transferred to a polyvinylidene fluoride membrane (Millipore, VT, USA). Nonspecific binding was blocked with 5% skim milk (BD Bioscience, NJ, USA) for 2 h, and the membrane was incubated with primary antibodies (rabbit anti-claudin-5 monoclonal antibody (1:5000), rabbit anti-occludin monoclonal antibody (1:5000), rabbit anti-VE-cadherin monoclonal antibody (1:5000), rabbit anti-β-catenin monoclonal antibody (1:5000), and rabbit anti-ZO-1 monoclonal antibody (1:5000) were purchased from Abcam (Cambridge, UK), and mouse anti-glyceraldehyde 3-phosphate dehydrogenase (GAPDH) monoclonal antibody (1:5000) was from Beyotime Biotechnology (Shanghai, China)) at 4 °C overnight. The membrane was washed four times with Tris-buffered saline-tween 20 (TBST) and then incubated with the respective secondary antibodies labeled with ‘horseradish peroxidase-conjugated secondary antibody’ (1:5000; ZSGB-Bio, Beijing, China) at room temperature for 2 h. The membrane was washed in TBST four times again and then color-rendered by an enhanced chemiluminescence reagent (Millipore, VT, USA) under ChemDoc XRS+ (Bio-Rad) software. Image Lab Software version 3.0 (Bio-Rad) was used to analyze the gray value of each strip. The relative expression levels of brain tissue proteins were expressed as the gray value of the target band over the gray value of GAPDH in the same sample. The relative protein expression levels in HBMECs were expressed as a ratio of the gray value of the target band to that of total proteins in the same sample [77].

### 4.7. Immunohistochemistry (IHC) and Immunofluorescence Staining Analysis

The expressions of claudin-5, occluding, and ZO-1 were detected by IHC analysis. Briefly, the rat brain was fixed in 4% paraformaldehyde for 20 min at 37 °C. Rat brain sections were incubated with claudin-5 (1:400), occludin (1:700), and ZO-1 (1:500) primary antibodies at 4 °C overnight. Subsequently, the sections were incubated with HRP-conjugated anti-rabbit/goat IgG at 37 °C for 20 min and color was then rendered by the diaminobenzidine (DAB) Immunohistochemistry Color Development Kit. IHC Images were visualized using Leica DMI6000B fluorescence microscopy with LAS AF LITE image processing software (Leica, Weztlar, Germany). HBMECs were stained with phalloidin (1:200, Yeasen Biotech Company, Shanghai, China) and DAPI to show F-actin and cell nucleus, respectively [75]. Finally, the images were gained using an N-STORM super-resolution microscope (Nikon, Tokyo, Japan).

### 4.8. Proteomics

#### 4.8.1. Protein Extraction and In-Gel Digestion

The rat brain (0.5 g) was put into 2 mL ice-cold 10 mM PBS (containing cocktail protease inhibitor; Roche Diagnostics, IN, USA). The mixture was homogenized thoroughly using a Teflon-glass homogenizer and then followed by 1 min sonication (1 s sonication and 2 s pause in ice-bath). The homogenate was then centrifuged at 12,000× *g* for 15 min at 4 °C, and the supernatant was collected. The protein concentration was determined by the Bradford method. In the CON and SMG groups, equal amounts of proteins from nine brain samples were mixed for one sample and mixed sample was loaded onto an SDS-PAGE system in triplicate.

After denaturation by a protein loading buffer (Solarbio), 50 μg protein was loaded onto an SDS-PAGE system (5% stacking gel + 12% separating gel) in triplicate. Electrophoresis was performed at 80 V for 30 min and 110 V for 80 min. The separating gel was stained by Coomassie blue dye and destained by 20% methanol. The stained gel was cut into strips along the lane after separation, and each lane was cut into four parts. In-gel digestion was mainly chopped for three steps: reduction, alkylation, and trypsin digestion. First, the gel pieces were rehydrated in 10 mM dithiothreitol at 60 °C for 30 min. Then, the liquid was removed, and acetonitrile was added into tubes for the dehydration of gel chips. Second, 200 μL of 2-iodoacetamide solution were added into tubes at room temperature for 30 min to be sufficiently alkylated. After reduction and alkylation, 12.5 ng/μL trypsin (Promega, WI, USA) was used for digestion at 37 °C overnight. Gel chips were mixed with 0.5% acetic acid and acetonitrile and incubated together for 15 min on a shaker incubator at 37 °C. Finally, dry peptide powder was obtained by rapid vacuum concentration and stored at −20 °C.

#### 4.8.2. High-Performance Liquid Chromatography (HPLC)-Tandem Mass Spectrometry (MS/MS) Analysis

The peptides were dissolved in 2% acetonitrile/98% H_2_O/0.1% formic acid (*v*/*v*/*v*). The peptide solution (5 μL) was injected into a C_18_ reverse-phase column (3 μm, 150 mm × 75 μm; Eksigent Technologies, MA, USA) and separated by a 1D-Ultra nanoflow HPLC system (Eksigent Technologies) coupled with a Triple TOF 4600 MS (AB Sciex, MA, USA). Mobile phase A was 0.1% formic acid/water, and mobile phase B was 0.1% formic acid/acetonitrile. A linear gradient from 5% of mobile phase B to 35% of mobile phase B over 60 min was used to separate the peptides.

The MS was operated in the positive ionization mode, and the MS1 scan range was from 350 to 1800 m/z and 15 K at m/z 700 for MS2 (mass range: 100–1000 m/z). The transient time was set at 256 ms. The highest 10 precursor ions with charge ≥ 2 were selected and then the f dissociation of protonated molecular ions was performed in auto MS/MS mode. The maximum ion injection times for the survey scan was 20 ms, while the MS/MS scans were 60 ms. The ion intensity threshold value for both scans was programmed at 10^6^.

#### 4.8.3. Protein Identification and Bioinformatics Analysis

The mass data analysis was performed by Max Quant (version 1.5.2.8). Protein sequence analysis was carried out using UniProt Database (https://www.uniprot.org/, accessed on 31 June 2020). The parameters were set as follows: the mass tolerance of the first search and the main search was 0.07 Da and 0.006 Da, respectively, whereas the MS/MS was 40 ppm. The minimum number of peptides to consider a protein was identified as 1, and unique + razor peptides were used for protein grouping and quantification. Based on the reversed database, the peptide false discovery rate < 0.01 was set at both peptide and protein levels. Carbamidomethyl cysteine was set as the fixed modification, whereas methionine oxidation was set as the variable modification. Trypsin was selected as the digestive enzyme with two missed cleavages at most.

Proteins from the CON and SMG groups were identified with three biological replicates. The ratio of protein abundance in the SMG group to that in the CON group was defined as fold change. Differentially expressed proteins (DEPs) were selected using conditions as follows: *p* < 0.05 and the fold change cutoffs were set as >1.5 and <0.667 for upregulation and downregulation, respectively. Then, the list of DEPs was inputted into online software DAVID version 6.8 (https://david.ncifcrf.gov/home.jsp, accessed on 10 March 2021) to perform Gene Ontology (GO) and Kyoto Encyclopedia of Genes and Genomes (KEGG) analyses. Annotation clusters with *p* < 0.01 were considered significant. Then, the parameter fold enrichment or gene counts were used to rank the significant terms. FA kinase (FAK) protein expression was examined using Western blot to validate the results obtained from the proteomic analysis. The experimental procedure followed the methods in Section 4.6.

### 4.9. Rac1, Wave2, and Arp3 Expression and the Ratio of F-Actin to G-Actin

Rac1, Wave2, and Arp3 expression in rat brain and HBMEC samples were measured by Western blot analysis. The ratio of F-actin to G-actin in the rat brain and HBMECs was analyzed using an F-actin/G-actin in vivo assay kit (Cytoskeleton, CO, USA) according to the manufacturer’s protocol as described previously. 8-pCPT-20-O-methyl-cAMP (O-Me-cAMP) was used as a specific agonist for Rac1. HBMECs were divided into four groups: CON, SMG, CON + agonist, and SMG + agonist. For CON + agonist and SMG + agonist groups, HBMECs were incubated with 200 μmol/L O-Me-cAMP for 30 min. Then, SMG + agonist group SMG for 24 h. The CON + agonist group was treated at 1 G for 24 h.26 Four groups of HBMEC samples were collected, and Rac1, Wave2, Arp3, F-actin, and G-actin expression was detected.

### 4.10. Statistical Analysis

Statistical analysis was performed using SPSS 20.0 software (IBM, New York, USA), and the results were expressed as mean ± standard deviation. The difference between the two groups was determined by one-way analysis of variance, and a *p*-value of < 0.05 was considered to indicate statistical significance.

## 5. Conclusions

This study suggested that SMG could disturb the integrity of BBB via induced inflammatory injury, damaged BBB ultrastructure, increased oxidative stress levels, increased permeability, and downregulated TJs and AJs protein expression in the rat brain. Further studies on HMBECs demonstrated that SMG induced HBMEC damage. SMG increased oxidative stress levels, promoted apoptosis, arrested the cell cycle phase, downregulated TJs and AJs protein expression, and altered F-actin distribution. These findings first revealed that the Rac1/Wave2/Arp3 pathway related to cell–cell junction and cell–ECM adhesion might be the potential mechanism for the damaged integrity of BBB under SMG. These results would help understand the change in BBB integrity when exposed to the MG environment and gain more insight into keeping the CNS homeostasis of astronauts during spaceflight.

## Figures and Tables

**Figure 1 ijms-22-05165-f001:**
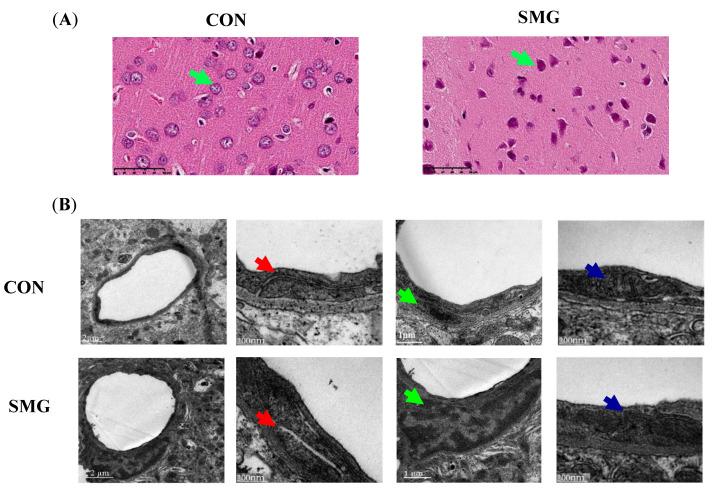
SMG induced inflammatory infiltration and damaged ultrastructure structure of BBB. (**A**): H&E staining in the cerebral cortex of CON and SMG groups, characterized by distinct infiltration of inflammatory cells (green arrows, scale bar, 250 µm). (**B**): Ultrastructure of BBB in the cerebral cortex as observed by transmission electron microscopy in CON and SMG groups. SMG group indicated that intercellular space was widened. Meanwhile, swollen pericytes and unclear mitochondria cristae were observed in the SMG group. Red arrows point to intercellular space, blue arrows point to intracellular mitochondria, scale bar, 200 nm, and green arrows point to pericyte, scale bar, 1 µm.

**Figure 2 ijms-22-05165-f002:**
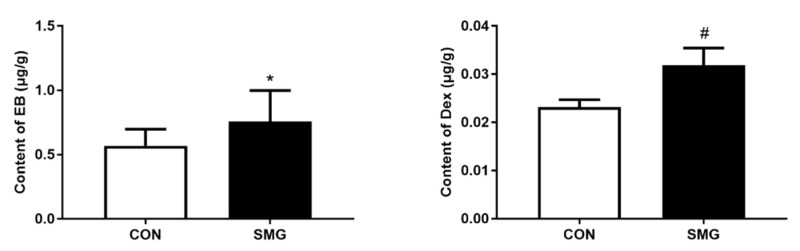
SMG induced BBB hyperpermeability in rats. Effects of SMG on the contents of EB and Dex in rats’ brains in CON and SMG groups. Note: Compared with the CON group, * *p* < 0.05, ^#^
*p* < 0.01.

**Figure 3 ijms-22-05165-f003:**
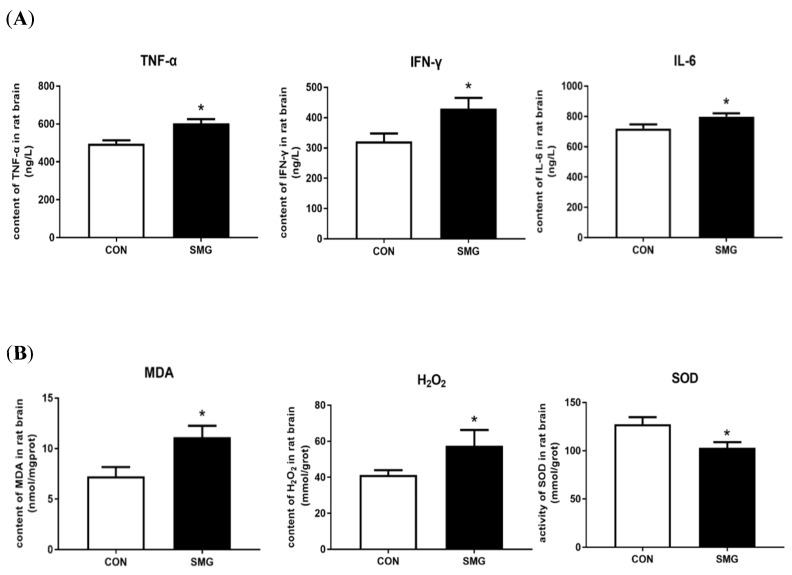
SMG increased oxidative stress and proinflammatory cytokine levels in rat brains. (**A**): The levels of TNF-α, INF-γ, and IL-6 in rat brains of the SMG group were significantly increased. (**B**): The content of MDA and H_2_O_2_ in the SMG group was remarkably increased. The SOD activity was decreased in the SMG group. Note: Compared with the CON group, * *p* < 0.05.

**Figure 4 ijms-22-05165-f004:**
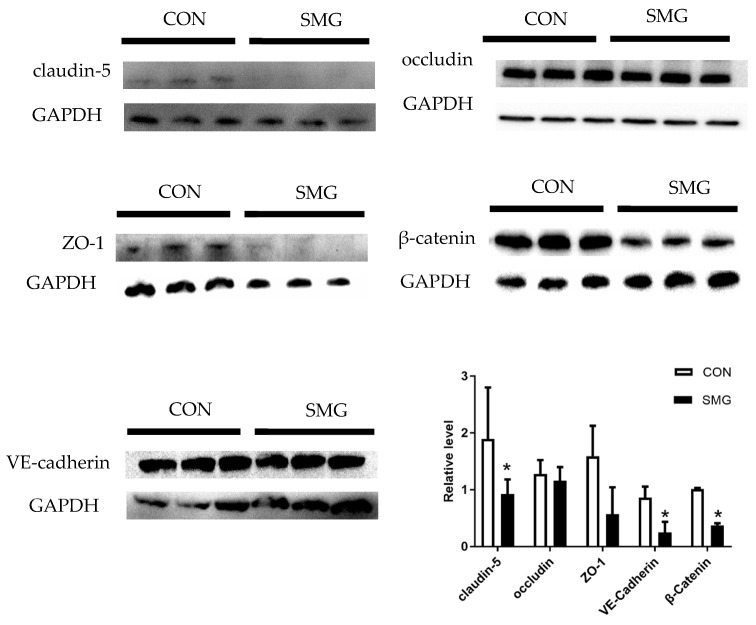
SMG dramatically decreased the expressions of claudin-5, VE-cadherin, and β-catenin in rat brains. Effects of SMG on protein expression of claudin-5, occludin, ZO-1, β-catenin, and VE-cadherin in rat brains as determined by Western blot. Note: Compared with the CON group, * *p* < 0.05.

**Figure 5 ijms-22-05165-f005:**
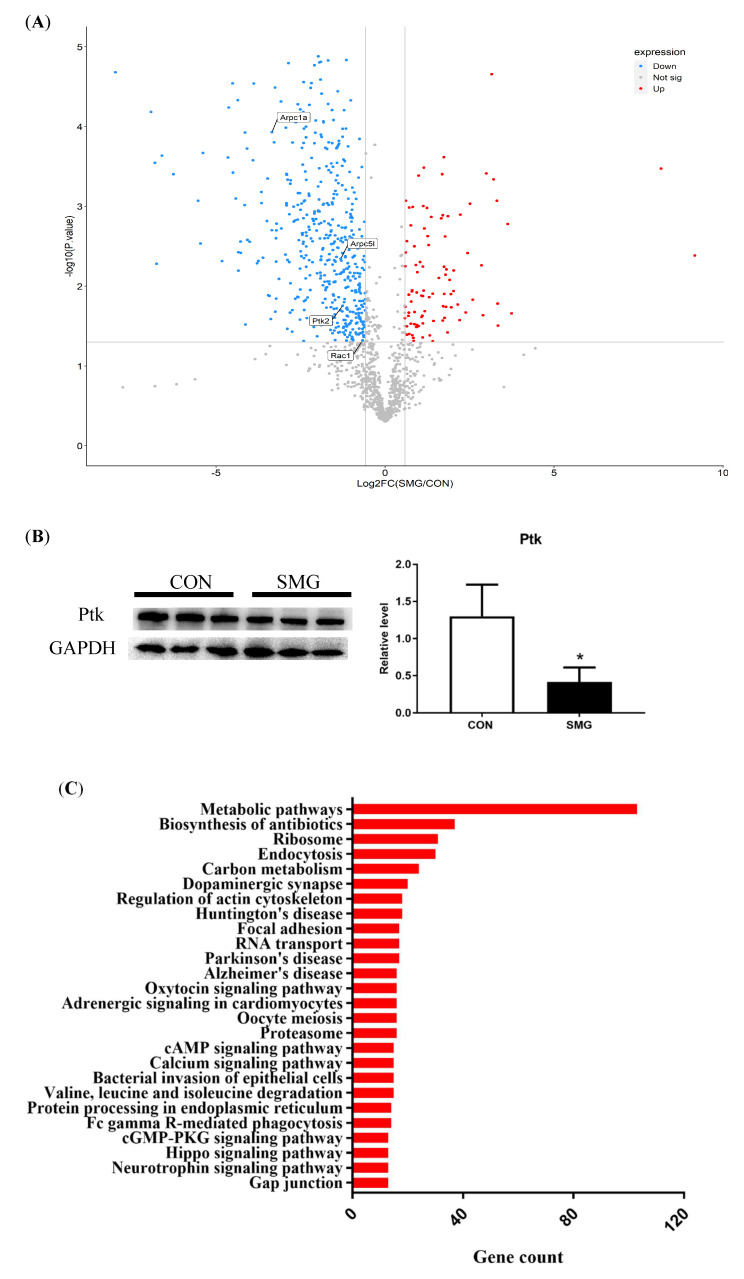
The result of proteomics analysis. (**A**): Volcano plot of all identified proteins. (**B**): Western blot of Ptk in rat brain under SMG for verifying the result of mass spectrometry. (**C**): KEGG pathway analysis of DEPs. Note: Compared with the CON group, * *p* < 0.05.

**Figure 6 ijms-22-05165-f006:**
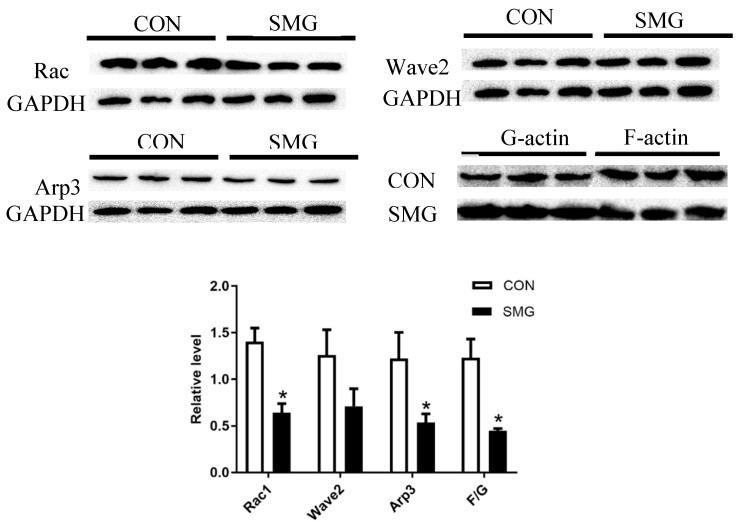
SMG inhibited the Rac1/Wave2/Arp2/3 pathway. Effects of SMG on protein expression of Rac1, Wave2, Arp3, F-actin, and G-actin in rat brains as determined by Western blot. Note: Compared with the CON group, * *p* < 0.05.

**Figure 7 ijms-22-05165-f007:**
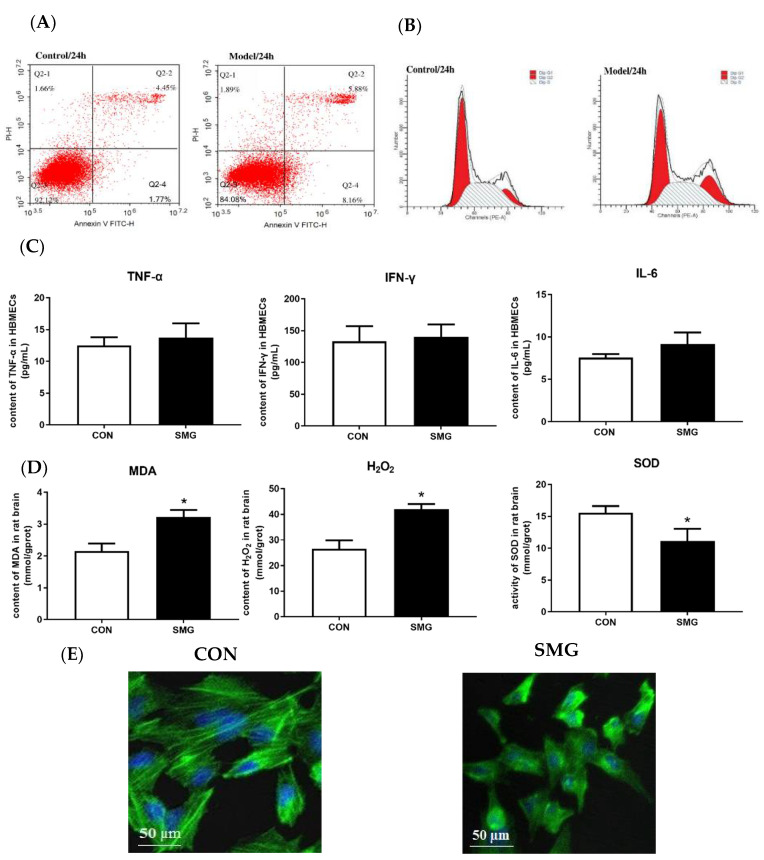
SMG induced apoptosis and oxidative stress injury and altered F-actin distribution in HBMECs. (**A**): Apoptosis in HBMECs under SMG (24 h). (**B**): Effects of SMG on the cell cycle of HBMECs. (**C**): The MDA content and H_2_O_2_ content were increased remarkably, and SOD activity decreased in the SMG group. (**D**), The proinflammatory cytokine (IL-6, IFN-γ, and TNF-α) levels in HBMECs were upregulated. (**E**), SMG altered the distribution of F-actin myofilaments in HBMECs (scale bar, 50 µm). Note: Compared with the CON group, * *p* < 0.05.

**Figure 8 ijms-22-05165-f008:**
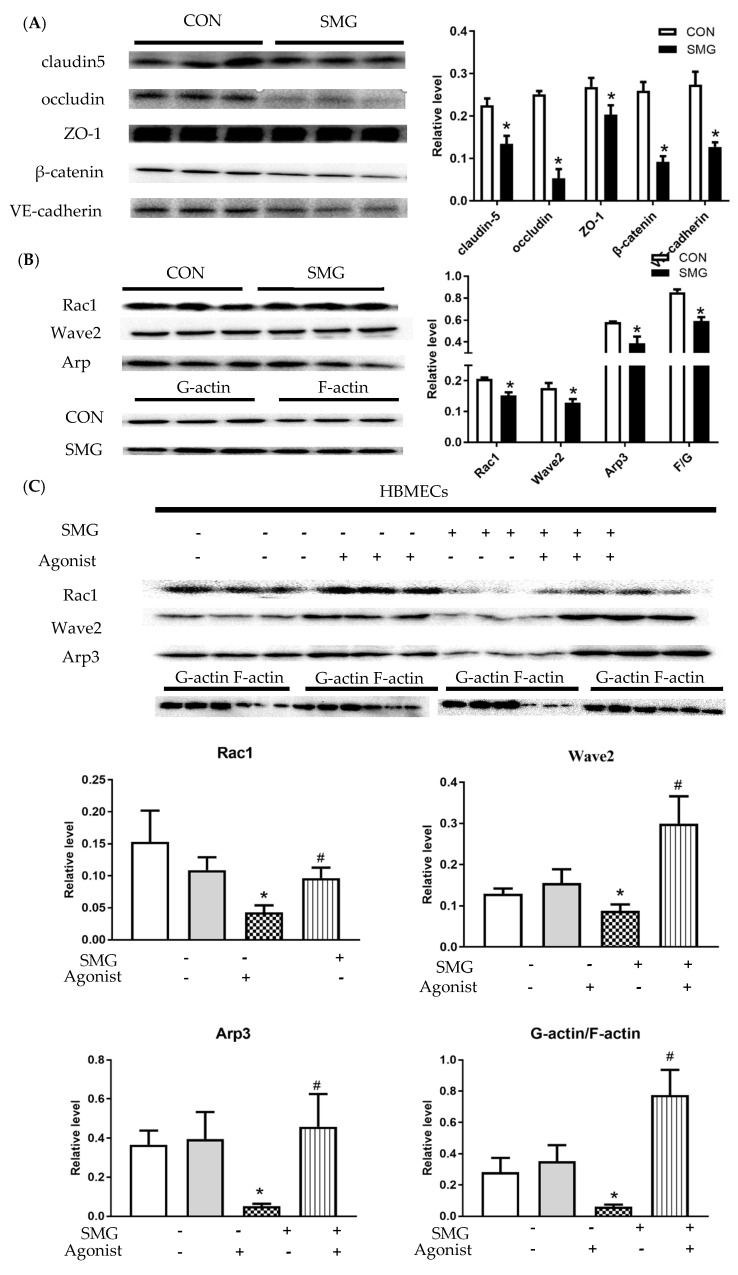
SMG downregulated the expression of AJs and TJs proteins and Rac1/Wave2/Arp3 pathway in HBMECs. (**A**), Effects of SMG on protein expression of claudin-5, occludin, ZO-1, β-catenin and, VE-cadherin in HBMECs as determined by Western blot. (**B**), The expression of Rac1, Wave2, Arp3, F-actin, and G-actin in HBMECs. (**C**), HBMECs were pretreated with Rac1 agonist for 30 min and then SMG for 24 h, and the expression of Rac1, Wave2, Arp3, F-actin, and G-actin were detected by Western blot. The relative protein expression levels in HBMECs were expressed as a ratio of the gray value of the target band to that of total proteins in the same sample. Respective bands with total proteins as loading control were shown in Appendix A. Note: Compared with the CON group, * *p* < 0.05, compared with the SMG group, ^#^
*p* < 0.05.

**Table 1 ijms-22-05165-t001:** Annotation clusters of all differently expressed proteins with enrichment scores (FE: fold enrichment).

Category	Term	Gene Counts	FE
Annotation Cluster 1	Enrichment Score: 13.61		
GOTERM_CC	cell–cell adherent junction	36	4.7
GOTERM_MF	cadherin binding involved in cell–cell adhesion	35	4.9
GOTERM_BP	cell–cell adhesion	34	5.2
Annotation Cluster 2	Enrichment Score: 11.66		
GOTERM_MF	GTP binding	51	4.2
GOTERM_MF	GTPase activity	32	5.0
GOTERM_MF	GDP binding	18	9.3
GOTERM_BP	small GTPase mediated signal transduction	23	3.2
Annotation Cluster 3	Enrichment Score: 7.6		
GOTERM_CC	proteasome complex	18	10.3
GOTERM_CC	proteasome core complex	11	17.6
GOTERM_MF	threonine-type endopeptidase activity	11	14.3
GOTERM_CC	proteasome core complex, alpha-subunit complex	5	18.7
GOTERM_BP	proteolysis involved in cellular protein catabolic process	8	5.1
Annotation Cluster 4	Enrichment Score: 5.51		
GOTERM_BP	translation	33	2.8
GOTERM_CC	cytosolic small ribosomal subunit	15	4.7
GOTERM_MF	structural constituent of ribosome	35	2.6
GOTERM_CC	ribosome	17	3.8
GOTERM_BP	cytosolic large ribosomal subunit	17	3.3

**Table 2 ijms-22-05165-t002:** Differently expressed proteins involved in cell adhesion.

UniProt IDs	Protein Names	Fold Change	*p*-Value
Q7TP36	shroom family member 2 (Shroom2)	12.369	0.0017
Q9Z1Z3	epsin 2 (Epn2)	9.2441	0.0005
O35964	SH3 domain-containing GRB2-like 1 (Sh3gl1)	3.4588	0.0072
Q5U2U2	v-crk avian sarcoma virus CT10 oncogene homolog-like (Crkl)	2.6584	0.0490
Q9JHL4	drebrin-like (Dbnl)	1.9196	0.0306
Q6AXS5	Serpine1 mRNA binding protein 1 (Serbp1)	1.9004	0.0325
A0JPJ7	Obg-like ATPase 1 (Ola1)	0.6494	0.0063
Q9JK11	reticulon 4(Rtn4)	0.6290	0.0282
P41562	isocitrate dehydrogenase (NADP(+)) 1, cytosolic (Idh1)	0.5536	0.0186
Q99N27	sorting nexin 1 (Snx1)	0.5484	0.0382
P61314	ribosomal protein L15 (Rpl15)	0.5042	0.0020
Q9QYL8	lysophospholipase II (Lypla2)	0.4824	0.0402
P04642	lactate dehydrogenase A (Ldha)	0.4518	0.0060
O88767	Parkinsonism associated deglycase (Park7)	0.4374	0.0002
P85972	vinculin (Vcl)	0.4353	0.0145
Q5PPJ9	SH3 domain-containing GRB2-like endophilin B2 (Sh3glb2)	0.4270	0.0066
P85845	fascin actin-bundling protein 1 (Fscn1)	0.4080	0.0048
Q5XFX0	transgelin 2 (Tagln2)	0.3826	0.0005
P63102	tyrosine3/5-monooxygenase activation protein, zeta (Ywhaz)	0.3470	0.00008
Q6JE36	N-myc downstream regulated 1 (Ndrg1)	0.3191	0.0094
Q6GMN2	Brain-specific angiogenesis inhibitor 1-associated protein 2 (Baiap2)	0.3057	0.0048
Q9Z269	VAMP associated protein B and C(Vapb)	0.2950	0.0112
Q5EGY4	YKT6 v-SNARE homolog (*S. cerevisiae*) (Ykt6)	0.2830	0.00008
P35281	RAB10, member RAS oncogene family (Rab10)	0.2583	0.0008
P35213	tyrosine 3/5-monooxygenase activation protein, beta (Ywhab)	0.1981	0.0057
Q9WTT7	basic leucine zipper and W2 domains 2(Bzw2)	0.1896	0.01543
Q4FZT2	protein phosphatase methylesterase 1 (Ppme1)	0.1874	0.0014
Q91Y81	septin 2 (Sept2)	0.1820	0.0022
O35509	RAB11B, member RAS oncogene family (Rab11b)	0.1397	0.0050
P62260	tyrosine3/5-monooxygenase activation protein, epsilon (Ywhae)	0.1304	0.0001
Q5I0D1	glyoxalase domain containing 4 (Glod4)	0.1077	0.0016
Q6NYB7	RAB1A, member RAS oncogene family (Rab1a)	0.0591	0.0002
Q568Z6	IST1, ESCRT-III associated factor (Ist1)	0.0560	0.0010
B1H267	sorting nexin 5 (Snx5)	0.0494	0.0038
Q66HA5	coiled-coil and C2 domain containing 1A (Cc2d1a)	0.0348	0.0132
Q07205	eukaryotic translation initiation factor 5 (Eif5)	0.0286	0.0138

**Table 3 ijms-22-05165-t003:** Differently expressed proteins involved in the regulation of actin cytoskeleton and bacterial invasion of epithelial cell.

UniProt IDs	Protein Names	Fold Change	*p*-Value
**Involved in** **regulation of actin cytoskeleton**			
O88377	phosphatidylinositol-5-phosphate 4-kinase type 2 beta (Pip4k2b)	4.4439	0.0170
P20171	GTPase HRas	0.6631	0.0122
P62138	protein phosphatase 1 catalytic subunit alpha (Ppp1ca)	0.5881	0.0253
P35465	p21 (RAC1) activated kinase 1 (Pak1)	0.5060	0.0462
P18666	myosin light chain 12B (Myl12b)	0.4906	0.0014
P45592	cofilin 1 (Cfl1)	0.4314	0.0001
Q9R0I8	phosphatidylinositol-5-phosphate 4-kinase type 2 alpha (Pip4k2a)	0.3561	0.0248
P21708	Mitogen-activated protein kinase 3 (Mapk3)	0.3446	0.0127
Q6GMN2	Brain-specific angiogenesis inhibitor 1-associated protein 2 (Baiap2)	0.3057	0.0048
P04937	fibronectin 1 (Fn1)	0.1998	0.0023
Q01986	mitogen activated protein kinase kinase 1 (Map2k1)	0.1986	0.0032
P63088	protein phosphatase 1 catalytic subunit gamma (Ppp1cc)	0.0429	0.0010
P85970	actin related protein 2/3 complex, subunit 2 (Arpc2)	0.0130	0.0004
**Involved in both regulation of actin cytoskeleton and bacterial invasion of epithelial cell**			
Q5U2U2	v-crk avian sarcoma virus CT10 oncogene homolog-like (Crkl)	2.6583	0.0489
Q6RUV5	ras-related C3 botulinum toxin substrate 1(Rac1)	0.6302	0.0478
P85972	vinculin (Vcl)	0.4353	0.0145
O35346	protein tyrosine kinase 2 (Ptk2)	0.4182	0.0176
A1L108	actin related protein 2/3 complex, subunit 5-like (Arpc5l)	0.4027	0.0045
Q99PD4	actin-related protein 2/3 complex subunit 1A (Arpc1a)	0.0981	0.0001
**Involved in bacterial invasion of epithelial cell**			
Q66HL2	Cortactin (Cttn)	2.2129	0.0003
B0BNF1	septin 8 (Sept8)	1.7935	0.0485
Q08877	dynamin 3 (Dnm3)	0.5656	0.0287
B3GNI6	septin 11 (Sept11)	0.5343	0.0069
P08082	clathrin, light chain B (Cltb)	0.3047	0.0009
Q9WU34	septin 3 (Sept3)	0.2932	0.0002
P04937	fibronectin 1 (Fn1)	0.1998	0.0023
Q91Y81	septin 2 (Sept2)	0.1820	0.0022

## Data Availability

Not applicable.

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
