# Peer review of "Rac1/Wave2/Arp3 Pathway Mediates Rat Blood-Brain Barrier Dysfunction under Simulated Microgravity Based on Proteomics Strategy"

_ijms, 2021, doi:10.3390/ijms22105165_

Round 1
Reviewer 1 Report
The authors present proteomics results using IHC, western blot, and mass spectrometry to study stimulated microgravity effects on the blood brain barrier. The authors found evidence of changing protein expression between controls and SMG in the Rac1/Wave2/Arp3 pathway, which could be a potential signaling pathway for blood brain barrier dysfunction.
Comments:
- In general the IHC figures are difficult to view.
- Figure 4 A the axis labels and legend is unreadable
- Page 4, line 129: Please indicate what data is being compared. Is this the IHC data?
- For the H&E experiments, how many samples were used?
- Page 5, line 171: How many total proteins were identified in the mass spectrometry experiment?
- For the mass spectrometry data: How many samples were analyzed?
The methods state on page 15, line 539; “equal amounts of proteins from three brain samples were mixed for one sample, resulting in three biological replicates in each group.” This is confusing. Were three samples mixed together for one sample, but repeated two more times for a three biological replicates?
- How was the mass spectrometry data normalized? How did you process the missing values? The table 2 infinity and 0 fold change values indicate zeros are still present in the data.
Did you use the LFQ or iBAQ normalization methods from MaxQuant? Did you apply a multiple test correction for the p-values, such as FDR?
- It would be helpful to label the proteins Rac1, Wave2, Arps (page 8, line 250), and FAK on the MS Volcano plot (Figure 4A). These proteins are not listed in table 2 but significant for the manuscript. Highlighting their p-value and fold change on the volcano plot will be helpful to evaluate the significance of these proteins.
9. Page 5, line 190: “With P<0.01, GO analysis generated 10 clusters …” Is this P value from the mass spectrometry data significance threshold? This is not the GO terms p-values?
10. Table 2. The column header “Gene IDs” should be changed to “UniProt IDs”. These are not Entrez Gene IDs.
- Page 11, line 330: “Our results showed that SMG downregulated claudin-5, VE-cadherin, and B-catenin expression. Claudin-5, VE-cadherin, and B-catenin downregulation might impair ..” change to
“Our results showed that SMG downregulated claudin-5, VE-cadherin, and B-catenin expression, which might impair ..”
- page 11, line 338: “inflammatory factors level have been increased…” change to “inflammatory factors were increased”.
- Page 11, line 341: Is the “downregulated to 0.06 to 0.43 times of the CON group” supposed to be “downregulated from 0.06 to 0.43 times” ? This sentence is confusing.
Author Response
Thank you again for your comments and we look forward to hearing from you regarding our submission. We would be glad to respond to any further question and comments that you have.
Please see the attachment for a point-by-point response.

Reviewer 2 Report
The main factor affecting astronauts is microgravity. Importantly, there are no mechanisms of adaptation to microgravity, unlike, for example, to reduction of partial oxygen pressure. A direct effect of microgravity is redistribution of fluid to the head, which leads to increased renal filtration, bone depletion, loss of muscle mass, changes in the immune and cardiovascular systems, changes in incoming sensory information. It would be interesting to assess the changes that occur under the combined action of the factors of space flight.
Author Response
Thank you again for your comments .
Reviewer 3 Report
The manuscript entitled “Rac1/Wave2/Arp3 Pathway Mediates Rat Blood-Brain Barrier Dysfunction under Simulated Microgravity based on Prote-3 omics Strategy” by Yan et al. claims that simulated microgravity (SMG) induces disruption of the blood brain barrier (BBB). They further claim that Rac1/Wave2/Arp3 could be a potential signaling pathway responsible for BBB disruption under SMG, based on the microscopic observations and proteomics analysis. However, there are several nonnegligible concerns, such as citation, logic, presentation of images, and description of methods and results, that prevent readers from accepting the authors’ claims.
Line 41. “It is worth noting that cognitive deficits and sleep disorders induced by MG are associated with blood-brain barrier (BBB) dysfunction, including increased permeability, and downregulated tight junctions (TJs) and adherens junctions (AJs) protein expression [8-10].” It seems that the references 8-10 do not support the authors’ claim that cognitive deficits and sleep disorders induced by MG are associated with BBB dysfunction. This statement is misleading that cognitive deficits in the microgravity condition is involved in BBB dysfunction. Please modify the text.
Line 54. “The regulation of the cytoskeleton is involved in the change of BBB permeability [15].” Ref. 15 is a publisher correction and therefore not appropriate as a citation.
Line 61. “Another study has shown that matrix metalloproteinase-9 (MMP-9) and aquaporin 4 expression was greatly exacerbated after hindlimb unloading in combination with low-dose γ-radiation, which would potentially cause BBB dysfunction [17].” There are two concerns in this statement. First, the word “exacerbate” is not usually used for changes in gene expression, because it is neither good nor bad itself. Second, although the authors’ study focuses only on MG but not on radiation, the authors cite literature that discusses the interaction between MG and radiation, which is not logical. Authors are expected to address these concerns.
Line 64. “It is implied that BBB could be dysfunctional under MG condition.” This statement does not sound logical because of the concern in the previous sentence described above.
Line 78. “The potential signaling pathway would be validated in rat and human BMEC (HBMEC).” Does it mean the pathway “was” validated in BMECs?
Line 80. “The expected finding might provide new insight on the deterioration of CNS homeostasis during space missions.” There is a concern on the logic in this statement because there is no direct relationship between the BBB function and the CNS homeostasis. In other words, even if the BBB was disrupted, that may not directly affect the homeostasis of the CNS.
Line 84. “Compared to the histomorphological image of CON, the SMG group showed signs of inflammatory cellular infiltration and nuclear pyknosis in the cerebral cortex (green arrows).” There is no green arrow in Figure 1. Moreover, due to the low magnification and resolution of the images, the infiltration of inflammatory cells claimed by the authors cannot be confirmed.
Line 89. “Meanwhile, there were observed swollen pericytes (green arrows) and unclear mitochondria cristae (blue arrows).” Again, the resolution is not enough to recognize the change in the morphology of mitochondria cristae.
Line 90. “The H&E staining results suggested that SMG induced inflammatory injury in the rat cerebral cortex. An inflammatory injury would increase BBB permeability, by modifying the localization of TJs proteins and downregulating the expression of TJs proteins [19]. … [20]. Available study has founded that … [21]. In addition, the destruction of mitochondria cristae revealed that SMG might have a change of oxidative stress levels and spontaneously induce BBB dysfunction [22]. Collectively, these results indicated that 21d-SMG damages brain histomorphology and BBB ultrastructure.” Basically, only the objective facts obtained from the experiment are mentioned in the Results section, and the reference-based discussion is done in the Discussion section. In other words, “suggested”, “would”, and “might” are not used in Results. The same principle should be applied for the other sections as well.
Line 108. “green arrows point to pericyte, scale bar, 1 μm” The scale bar is hardly visible. Figs 7A, B, C, and D are missing. Fig 7E does not have a scale bar. Figs 8A and B are missing. The definitions of the symbols “*” and “#” are unclear. Moreover, the method of statistical analysis in the two-group comparison (Fig. 8A?) should be different from that in the four-group comparison (Fig. 8B?), while there is a description about only “analysis of variance” (line 607). The method for statistical analysis for four-groups experiment should be defined carefully, because it determines the credibility of the interpretation of data.
Line 486. “The hydrogen peroxide (H2O2) and malondialdehyde (MDA) levels and superoxide dismutase (SOD) activity in rat brain tissue and cell samples were determined using a commercial kit according to the manufacturer’s instructions (Nanjing Jiancheng Bioengineering Institute, Nanjing, China).” The measurement principle that underlies the reliability of this experiment is unclear in this description. The product number (or name) should be indicated. A manual or a datasheet of the kit may be useful to specify the principle.
Line 608. “4.2. Animal treatment and sample collection” Why is it in the section of statistical analysis?
Author Response

(The authors gave the same response as above.)
